# Effects of Intracerebroventricular and Intra-Arcuate Nucleus Injection of Ghrelin on Pain Behavioral Responses and Met-Enkephalin and β-Endorphin Concentrations in the Periaqueductal Gray Area in Rats

**DOI:** 10.3390/ijms20102475

**Published:** 2019-05-19

**Authors:** Samaneh Pirzadeh, Javad Sajedianfard, Anna Maria Aloisi, Mahboobeh Ashrafi

**Affiliations:** 1Department of Basic Sciences, School of Veterinary Medicine, Shiraz University, Shiraz 71441-69155, Iran; samaneh_pirzadeh@yahoo.com (S.P.); mashrafi@shirazu.ac.ir (M.A.); 2Department of Medicine, Surgery and Neuroscience, University of Siena, 53100 Siena, Italy; annamaria.aloisi@unisi.it

**Keywords:** ghrelin, formalin test, met-enkephalin, beta-endorphin, microdialysis

## Abstract

Ghrelin is an endogenous ligand for orphan growth hormone secretagogue receptors. Ghrelin receptors have been found in central nervous system (CNS) areas responsible for pain modulation and transmission. This study investigated the effects of intracerebroventricular (ICV) and intra-arcuate nucleus (ARC) injection of ghrelin on pain behavioral responses and levels of β-endorphin (β-EP) and met-enkephalin (MENK) in the periaqueductal gray area (PAG) during the formalin test in rats. Thirty-five male rats were studied in five groups. Ghrelin was injected into the left lateral ventricle (ICV, 5 µL) or into the ARC (1 µL). After 15 min, formalin (2.5%) was subcutaneously injected into the left hind paw. Behavioral nociceptive scores were recorded for 60 min. MENK and β-EP were collected by microdialysis in the PAG and determined by high-performance liquid chromatography (HPLC). ICV and ARC injection of ghrelin significantly reduced pain in all phases of the formalin test (*p* < 0.001). Dialysate concentrations of MENK and β-EP in the PAG increased in all the phases (*p* < 0.01). In conclusion, the present study shows that the ARC nucleus and the endogenous opioid system are involved in ghrelin-induced pain modulation.

## 1. Introduction

Pain is an unpleasant sensory experience caused by intense or damaging stimuli and regulated by the consolidation of complex actions modulated by central factors [1]. Many parts of the central nervous system (CNS), such as the periaqueductal grey area (PAG), play an important role in pain modulation. β-endorphin (β-EP) and met-enkephalin (MENK) are released into the PAG and play an important role as endogenous opioids in the antinociceptive system [2].

Ghrelin, a 28-amino acid gastric acylated peptide, is an endogenous ligand for orphan growth hormone secretagogue receptors (GHS-R) [3,4]. Ghrelin is originally secreted by the stomach, but it is also found in several regions of the nervous system [3,5,6]; its receptors have been found in different areas of the CNS, including various hypothalamic nuclei such as the arcuate nucleus, other areas of the rat brain such as the hippocampal formation and thalamic regions, and in the spinal cord. Since many of these structures are responsible for controlling pain transmission [6,7,8,9], ghrelin and its receptors have been suggested to play a key role in the antinociceptive system [10]. Ghrelin has been shown to have an inhibitory effect on inflammatory pain through interference with the central opioid system [10,11]. Moreover, previous research has demonstrated that it has excitatory effects on the ventromedial arcuate nucleus (ARC), where neurons contain endogenous opioids and project to the PAG [9,12]. It has been observed that neurons containing ghrelin innervate other peptidergic systems such as neurons containing pro-opiomelanocortin (POMC) [13]. POMC-derived β-EP plays a key role in the antinociceptive system [14]. POMC and enkephalin neurons have been identified in ARC terminal fields in the PAG [15,16]. Indeed, the PAG is rich in opioid receptors, and β-endorphinergic fibers from the ARC extend to the PAG [12]. Enkephalinergic neurons that end up in the PAG are presumably derived from the ARC [16,17]. In agreement with these data, ghrelin was found to inhibit the inflammatory pain induced by carrageenan in rats via interaction with the central opioid system [18]. Other researchers have shown that ghrelin palliates chronic neuropathic pain [19,20] and prohibits cisplatin-induced mechanical hyperalgesia and cachexia [21].

In the present study, the effects of injection of ghrelin into the ARC or lateral ventricle (ICV) in rats were investigated in the formalin test, a well-known model of inflammatory pain [22]. In addition, levels of β-EP and MENK were collected by microdialysis in the PAG and measured by high-performance liquid chromatography (HPLC).

## 2. Results

### 2.1. Saline Injection Versus Sham Treatment 

To determine the possible effects of ICV vs. ARC injection of saline and sham treatment, one-way analysis of variance (ANOVA) was applied to the pain scores and β-EP and MENK concentrations with the factors Group (3 levels: Saline-ARC, Saline-ICV, SHAM) and Time (12 levels: 12 5-min periods). There were no significant differences among groups (Figure 1, *p* > 0.05) but only a significant effect of Time due to the different formalin phases (data not shown). Thus, the SHAM group was no longer considered.

### 2.2. Evaluation of the Effects of Ghrelin on the Pain Score after Intracerebroventricular (ICV) or Intra-Arcuate Nucleus (ARC) Injection 

To evaluate ghrelin’s effects in the two injection sites, analysis of variance (ANOVA) was carried out with the factors Treatment (2 levels: Saline, Ghrelin) and Time (12 levels: 12 5-min periods repeated) (Figure 2A,B). As reported in the Materials and Methods section, the volumes were different: ICV 5 μL, ARC 1 μL. 

ICV treatment: as shown in Figure 2A, ANOVA revealed significant effects of Treatment (F(1,9) = 452.64, *p* < 0.0001), Time (F(11,99) = 9.59, *p* < 0.0001) and the interaction Treatment × Time (F(11,99) = 4.12, *p* < 0.0001). This was due to the strong decrease of pain scores in the ghrelin-treated group (*p* < 0.001), confirming the analgesic effect of ghrelin. The decrease was present particularly in the first and intermediate phases of the formalin test; in the second phase, the changes were less marked (Figure 2A).

ARC treatment: as shown in Figure 2B, ANOVA revealed significant effects of Treatment (F(1,14) = 176.06, *p* < 0.0001), Time (F(11,154) = 10.92, *p* < 0.0001) and the interaction Treatment × Time (F(11,154) = 5.11, *p* < 0.0001). In animals injected in the ARC nucleus the effect was similar to that after the ICV injection. Indeed after a marked decrease in the first and intermediate phases of the formalin test, pain scores tended to recover during the second phase (Figure 2B).

### 2.3. Evaluation of the Effects of Ghrelin on β-Endorphin (β-EP) Levels in the Periaqueductal Grey Area (PAG) 

ANOVA was applied to the β-EP levels determined in the PAG with the factors Treatment (2 levels: Saline, Ghrelin) and Time (8 levels: S1–S8) (Figure 3A,B). 

ICV treatment: in this group of subjects there were significant effects of Treatment (F(1,8) = 81.9, *p* < 0.0001), Time (F(7,56) = 80.65, *p* < 0.001) and the interaction Treatment × Time (F(7,56) = 45.5, *p* < 0.001). As shown in Figure 3A, this was due to the higher β-EP levels in the Ghrelin group than in the Saline one from the third (S3) to the seventh (S7) samples, a time period that includes the complete duration of the formalin test (*p* < 0.01). 

ARC treatment: in this group of subjects there were significant effects of Treatment (F(1,8) = 94.1, *p* < 0.0001), Time (F(7,56) = 23.66, *p* < 0.001) and the interaction Treatment × Time (F(7,56) = 13.89, *p* < 0.001). This was due to the higher β-EP levels in the Ghrelin group than in the Saline one from S3 to S7 (Figure 3B).

### 2.4. Evaluation of the Effects of Ghrelin on Met-Enkephalin (MENK) Levels in the PAG 

ANOVA was applied to the MENK levels determined in the PAG with the factors Treatment (2 levels: Saline, Ghrelin) and Time (8 levels: S1–S8) (Figure 4A,B). 

ICV treatment: in this group of subjects there were significant effects of Treatment (F(1,8) = 140.9, *p* < 0.0001), Time (F(7,56) = 48.09, *p* < 0.001) and the interaction Treatment × Time (F(7,56) = 33.03, *p* < 0.001). As shown in Figure 4A, this was due to the higher MENK levels in the Ghrelin group than in the Saline one from S3 to S7 (*p* < 0.01). 

ARC treatment: in this group of subjects there were significant effects of Treatment (F(1,8) = 62.41, *p* < 0.0001), Time (F(7,56) = 79.83, *p* < 0.001) and the interaction Treatment × Time (F(7,56) = 31.84, *p* < 0.001). This was due to the higher MENK levels in the Ghrelin group than in the Saline one from S3 to S7 (Figure 4B).

## 3. Discussion

In this study, ICV and ARC injection of ghrelin significantly decreased the nociceptive scores in all three phases of the formalin test (first, intermediate, second). In parallel, ghrelin increased the concentrations of MENK and β-EP in the PAG, an effect that lasted throughout the formalin test. A summary of the neural centers involved in these actions is shown in Figure 5.

Recent studies have indicated that ghrelin has a role in controlling pain [23]. Ghrelin administration was shown to prevent the progress of hyperalgesia induced by intraplantar carrageenan injection in rats [18]. In the present experiment we demonstrate that the effect can be obtained with both ICV injection and a more specific injection of ghrelin into the ARC nucleus. 

Several studies reported that ghrelin and its receptors were expressed in various brain areas, such as the hypothalamus [6,8,11]. The location of ghrelin and its receptors is indicative of their role in the modulation of pain systems [10]. Ghrelin has stimulating effects on neurons of the ventromedial ARC [9], which contains endogenous opioid-containing neurons [12]; indeed, ghrelin activates pro-opiomelanocortin (POMC) neurons [24] and then increases the release or synthesis of opioid peptides [13]. POMC-derived β-endorphin plays an important role in the descending antinociceptive pathway [18,23]. Studies have shown that ghrelin modulates inhibitory transmission in the spinal cord dorsal horn, stops cisplatin, and induces mechanical hyperalgesia and cachexia [8,11,21]. Moreover, ghrelin increases the levels of hypothalamic nitric oxide (NO) synthase [25]. Neuronal NO modulates the antinociceptive effect of endogenous opioids by activating μ-opioid receptors [24,26]; therefore, ghrelin probably enhances the antinociceptive effects of endogenous opioids via the NO pathway.

It has been shown that neurons containing β-EP are sent out from the ARC to the nucleus accumbens and lateral septum, and then extend to the PAG, dorsal raphe nucleus and locus coeruleus [12,27]. The PAG has an important role in the descending pain control pathway [2]. It was the first brain region explicitly demonstrated to activate an endogenous pain inhibitory system. Early studies showed that microinjection of opioids or electrical stimulation applied to this region elicited a powerful antinociceptive effect in animals and humans [28,29,30,31]. The brain-stem structures send descending impulses to the spinal cord and inhibit the transmission of nociception signals in this region [32]. The β-EP nerve terminals in the PAG probably indicate the involvement of the ARC/PAG endorphinergic pathway; hence, β-EP is the neurotransmitter involved in the nucleus accumbens-to-PAG pathway [33]. Previous studies have shown that the PAG and raphe nucleus are rich in opiocortin–ir [17] and also opioid receptors [34]. Fibers of the ARC project to the PAG and raphe nucleus [16]. There are neurons immunoreactive to substance *p*, neurotensin and encephalin in the ARC that end up in the PAG [16]. Microinjection of morphine into the nucleus accumbens was blocked by MENK antibodies or naloxone injection, indicating a descending pain modulatory pathway from the nucleus accumbens to the PAG and endogenous opioids. MENK, in particular, functioned as a neurotransmitter [13].

Azizzadeh et al. [23] demonstrated that intraperitoneal administration of high doses of ghrelin significantly reduced the nociceptive score in both the early and second phases of formalin pain. Kutlu et al. [35] showed that ICV administration of ghrelin could decrease the acute pain threshold in mice. We found that focal injection of ghrelin could decrease the nociceptive score; the effects were clear and long-lasting in both experimental conditions, although they appeared to be more evident after the ICV injection. Thus, although we cannot carry out a direct comparison between the two injection sites, also due to the different volumes injected, we suggest that the ARC nucleus is the key point in the analgesic action of ghrelin. In the present experiment, the results of the ARC injection strongly suggest an involvement of this specific area in the ghrelin-induced effects. Moreover, our microdialysis data demonstrated that ghrelin injection increased the concentrations of MENK and β-EP in the PAG during the formalin test. This suggests that the analgesic effects seen in the formalin test were mediated by ghrelin receptors and then the opioidergic pathways in the brain. Naloxone, the opioid receptor antagonist, decreased tail withdrawal latency and paw-licking latency and antagonized the effect of ghrelin; co-administration of ghrelin and [D-lys3]-GHRP-6 (ICV) antagonized the antinociceptive effects of ghrelin [36]. Other data confirm the interaction between ghrelin and the opioid system. For instance, the ghrelin receptor agonist GHRP-2 increased the latency in the tail withdrawal test through interaction with GHS-R1α and then opioid receptors, co-administration of [D-lys3]-GHRP-6 antagonized the antinociceptive effects of GHRP-2, and ICV injection of the opioid receptor antagonist naloxone reversed the antinociceptive effects of GHRP-2 [37,38]. 

In conclusion, we have demonstrated that the ghrelin-induced analgesic effect can be attributed to the endorphinergic pathways running from the ARC to the PAG. It should be underlined that although a 5-fold lower amount of ghrelin was injected into the ARC than into the ventricle, we obtained comparable results. Thus, as suggested by the widespread distribution of ghrelin receptors in the CNS, the analgesic effect is not limited to the ARC, but we can indicate it as a very effective site for the action of ghrelin.

## 4. Materials and Methods 

### 4.1. Ethics and Animals 

The protocol used in this study was approved by the Ethics Committee of the School of Veterinary Medicine, Shiraz University, Shiraz, Iran (code 95GCU1M1293; 16/01/2017).

Thirty-five adult male Sprague–Dawley rats (280 ± 30 g) were used. For adaptation, they were maintained for one week in an animal room at 22 ± 2 °C and a 12/12 h light/dark cycle with food and water ad libitum.

### 4.2. Study Design 

Before being tested, all animals, except the sham subjects, were implanted with a guide cannula in the lateral ventricle (ICV) or in the arcuate nucleus (ARC) and another one in the periaqueductal gray matter (PAG) to allow microdialysis. Twenty-four hours after surgery, the microdialysis and formalin test were performed (see schema in Figure 6). Ghrelin (Sigma-Aldrich) was added to normal saline and injected ICV or ARC in the test groups. In the control groups, equal volumes of normal saline were used. The injection was performed with a Hamilton syringe (5 µL) for one minute.

The formalin test was started after the optimal time for the outcome of the drug (15 min). All groups received 50 μL formalin 2.5% subcutaneously in the left hind paw. Animals were then divided into five groups (n = 7 rats per group) depending on the drug and the site of injection:(1)Saline-ARC: receiving normal saline (1 μL) in the arcuate nucleus;(2)Saline-ICV: receiving normal saline (5 μL) ICV;(3)SHAM: receiving no injection;(4)Ghrelin-ICV: receiving 1 nmol ghrelin (5 μL) ICV;(5)Ghrelin-ARC: receiving 0.2 nmol ghrelin in the arcuate nucleus (1 μL).

### 4.3. Implantation of the Probe and Guide Cannula and Injection of Drugs 

About one week after arrival, rats were anesthetized with sodium pentobarbital (50 mg/kg i.p.). One guide cannula was then implanted vertically in the lateral ventricle (AP = −0.8 mm, L = +1.5 mm, DV = −3.6 mm) or in the arcuate nucleus (AP = −2.3 mm, L = +0.5 mm, and DV = −10 mm, and angle 15°) [39]. After implantation of the first cannula, a microdialysis probe was implanted in the PAG (AP = −7.6 mm, L = +0.6 mm, and DV = −5.8 mm). 

### 4.4. Microdialysis and Sample Collection 

During the microdialysis, ACSF (artificial cerebrospinal fluid: NaCl 114, CaCl_2_ 1, KCl 3, MgSO_4_ 2, NaH_2_PO_4_ 1.25, NaHCO_3_ 26, NaOH 1, glucose 10, pH = 7.4) was perfused by means of a microinjection pump into the microdialysis probe with a flow rate of 2.0 µL/min (WPI, SP 210, syringe pump). After 15 min, 30 μL of the perfused ACSF were collected in individual 1.5 mL sterile Eppendorf tubes located in dry ice. Eight samples were collected and immediately stored in a freezer at −80 °C until analysis. As shown in Figure 6, the first sample (S1) was a baseline sample without ghrelin and formalin effect; the second sample (S2) was a baseline sample without formalin effect but with ghrelin effect. The third sample (S3) was related to the acute and inter phases of the formalin test (0–15 min). The fourth, fifth and sixth samples (S4–S6) were related to the second phase of the formalin test, collected at 15–30, 30–45 and 45–60 min of the formalin test. S7 and S8 were samples after the formalin test.

### 4.5. Formalin Test

For induction of pain in all groups, 50 μL of 2.5% formalin was injected subcutaneously into the left hind paw with a 27-gauge needle 15 min after the ICV or ARC injections. The nociceptive score was recorded every 15 s for one hour following formalin injection: score 0, if the animal did not display any certain sign; score 1, if the animal’s left foot remained on the ground and it did not put pressure on it; score 2, if the animal tapped the ground with its foot or flexed the limb toward the abdomen; score 3, if the animal licked the injected paw [40]. The injection of formalin generates a biphasic response. The acute phase is the first 5 min after injection. The second phase starts from the 20th minute and ends at the 60th minute. Between the 5th and 20th minutes is the inter phase [2]. 

### 4.6. High-Performance Liquid Chromatography (HPLC) Analysis

Microdialysis samples were measured with a Knauer high-performance liquid chromatograph (HPLC) with ultraviolet (UV) detection and a Smart line pump (1000). The β-EP and MENK separation were carried out isocratically with a Knauer reverse phase C18 column (vertex plus 250 × 4.6 mm Eurospher100-5) with guard cartridge and a Knauer CTO-6A column oven. The system was used at oven temperature (40 °C) with detection at 210 nm. The flow rate was set at 0.2 mL/min to elute samples of the column. Solvents were degassed under vacuum. The buffer was KH_2_PO_4_ (68 mL) and acetonitrile (32 mL) 0.1 M (pH 2.3). The mobile phase was acetonitrile/buffer in the first 4 min, acetonitrile/water from 4 to 6 min, and acetonitrile/buffer from 6 to 12 min (temperature 40 °C). Plotting of graphs and data analyses were performed with Autochro data module software. The concentrations of each sample were calculated by interpolating the areas obtained after fraction injection with those obtained from three known standard concentrations (met-enk from Abcam and β-end from Sigma-Aldrich, dissolved in normal saline) injected regularly into the HPLC to monitor the linearity of the system. Integration of chromatographic data was performed with Perkin-Elmer 1020 software [41].

### 4.7. Histological Verification

For verification of the locations of the guide cannulae in the lateral ventricle and ARC and the microdialysis probe in the PAG, rats were euthanized with an overdose of diethyl ether at the end of the experiment and their brains were fixed with formalin. The correct position of the cannulae and probe tracing were verified in all the rats by comparison with the rat brain atlas [39].

### 4.8. Data Analysis

Data are reported as mean ± SEM in the figures. Analysis of variance (ANOVA repeated measure) was applied to the formalin-induced behavioral responses with the factors Treatment (2 levels: Saline, Ghrelin), Injection site (2 levels: ICV, ARC) and Time (repeated, 12 5-min periods). *p* < 0.05 was considered significant. Duncan’s multiple range test was used as the post hoc test. 

## 5. Conclusions

Ghrelin significantly decreased the nociceptive score in the formalin test and increased the MENK and β-EP concentrations in the PAG when injected in the lateral ventricle and in the arcuate nucleus of rats. The increase of opioids in the PAG is probably due to the action of ghrelin in the ARC, lateral ventricle or its related nuclei, and ghrelin probably acts via the activation of ghrelin receptors and then opioid receptors. The mechanisms underlying the greater effects of ICV injection of ghrelin on the PAG opioid levels require further study.

## Figures and Tables

**Figure 1 ijms-20-02475-f001:**
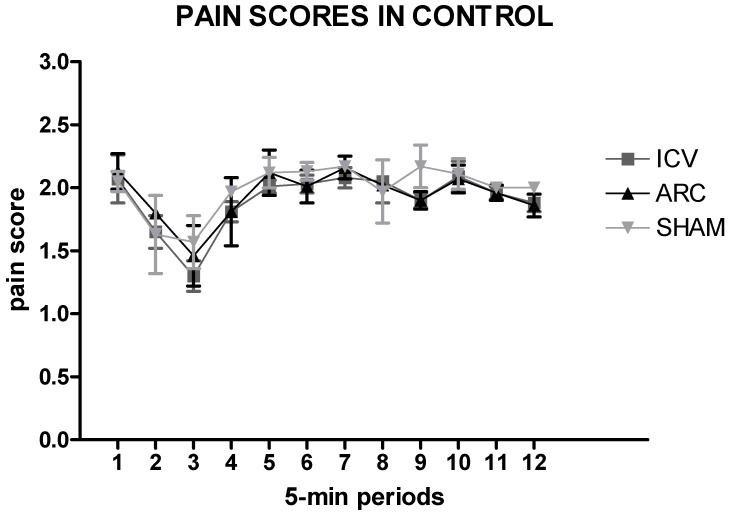
Time courses of the pain scores recorded in the control groups (Saline-lateral ventricle (ICV), Saline-ventromedial arcuate nucleus (ARC), SHAM) during the formalin test (60 min) divided into 5-min periods. Data are mean ± standard error of the mean (SEM).

**Figure 2 ijms-20-02475-f002:**
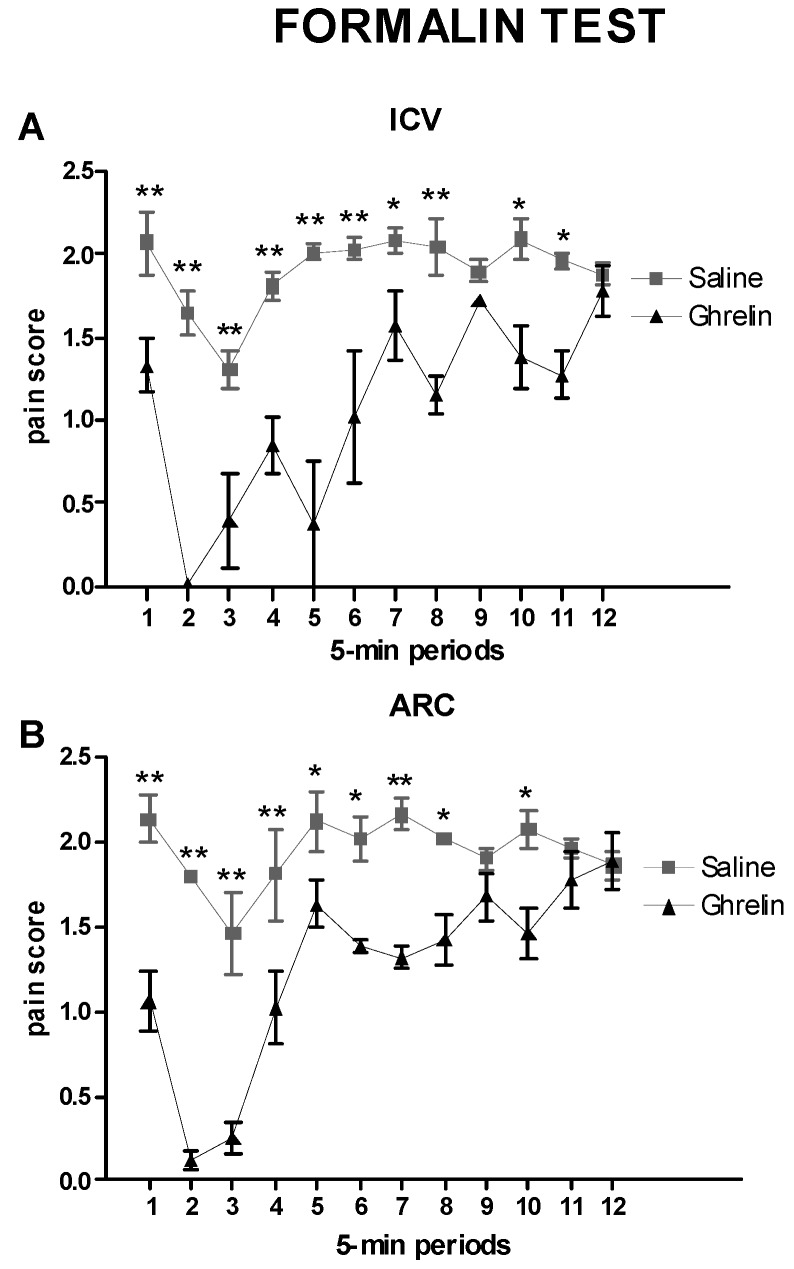
(**A**) Time courses of the pain scores recorded during the formalin test in animals treated with Saline-ICV or Ghrelin-ICV. (**B**) Time courses of the pain scores recorded during the formalin test in animals treated with Saline-ARC or Ghrelin-ARC. Data are mean ± SEM. ** *p* < 0.001 and * *p* < 0.05 vs. other group, same time period.

**Figure 3 ijms-20-02475-f003:**
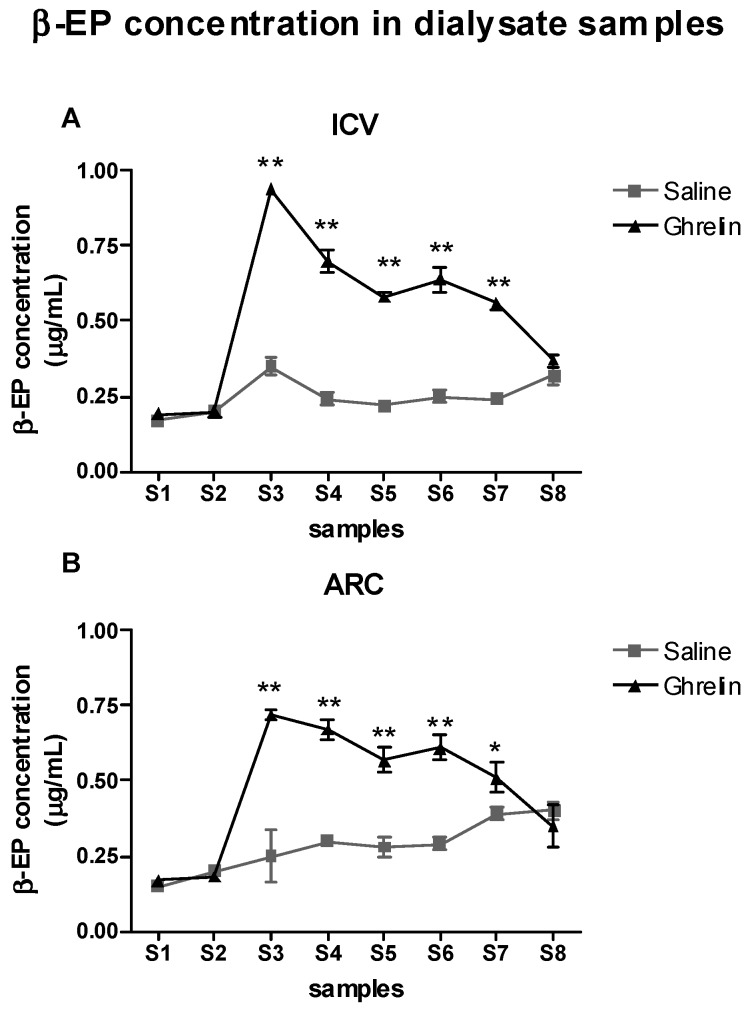
(**A**) Time courses of the β-endorphin (β-EP) dialysate levels determined before, during and after the formalin test in animals treated with Saline-ICV or Ghrelin-ICV. (**B**) Time courses of the β-EP dialysate levels determined before, during and after the formalin test in animals treated with Saline-ARC or Ghrelin-ARC. Data are mean ± SEM. ** *p* < 0.001 and * *p* < 0.05 vs. other group, same time period.

**Figure 4 ijms-20-02475-f004:**
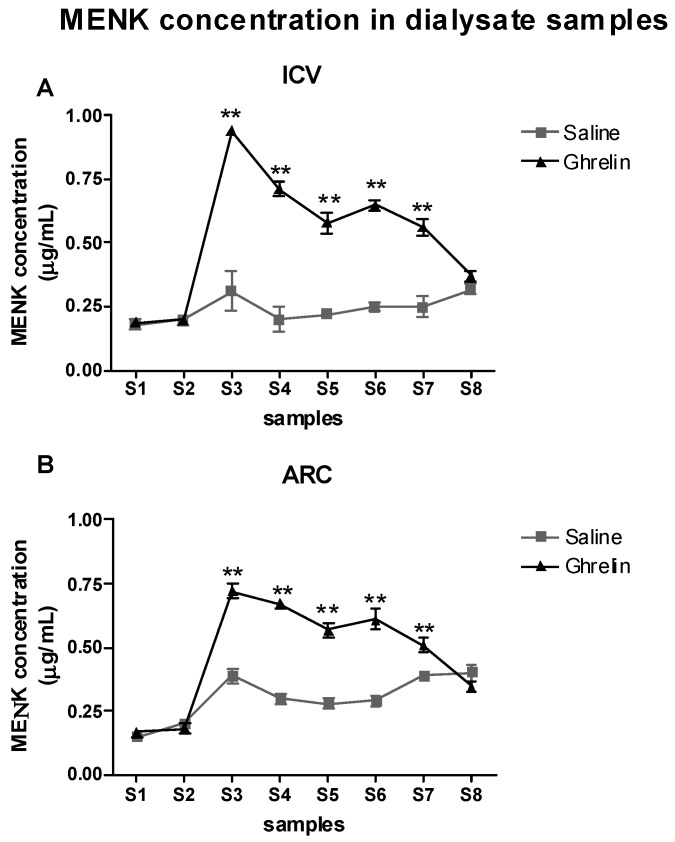
(**A**) Time courses of the met-enkephalin (MENK) dialysate levels determined before, during and after the formalin test in animals treated with Saline-ICV or Ghrelin-ICV. (**B**) Time courses of the MENK dialysate levels determined before, during and after the formalin test in animals treated with Saline-ARC or Ghrelin-ARC. Data are mean ± SEM. ** *p* < 0.001 and * *p* < 0.05 vs. other group, same time period.

**Figure 5 ijms-20-02475-f005:**
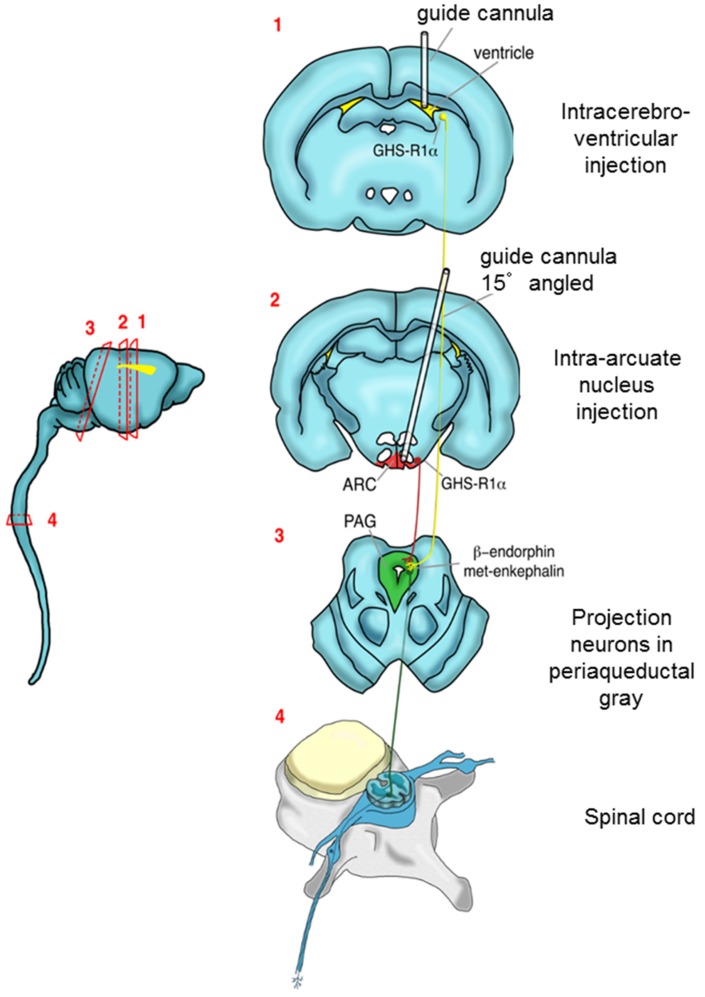
Graphical representation of the neural pathways considered in the study.

**Figure 6 ijms-20-02475-f006:**
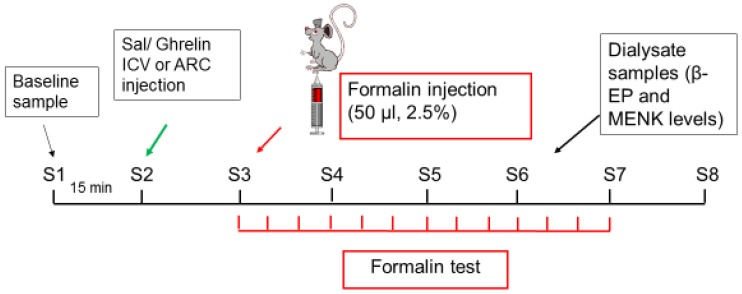
Schematic representation of the experimental design. The microdialysis and formalin test were performed 24 h after implantation of the microdialysis probe. Sample 1 (S1, 30 μL) was collected as the baseline value, and then Saline/Ghrelin were injected into the lateral ventricle (ICV) or arcuate nucleus (ARC) (S2). Dialysate samples were collected every 15 min for 2 h, the last sample (S8) being 15 min after the end of the formalin test. The formalin test was started after the optimal time for the outcome of the drug (15 min), and pain scores were recorded every 5 min for one hour.

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
