# Peer review of "Effects of Intracerebroventricular and Intra-Arcuate Nucleus Injection of Ghrelin on Pain Behavioral Responses and Met-Enkephalin and β-Endorphin Concentrations in the Periaqueductal Gray Area in Rats"

_ijms, 2019, doi:10.3390/ijms20102475_

Reviewer 1 Report

In this article, Pirzadeh et al were interested by the anti-nociceptive effects of Ghrelin and its central effects. They report the effects of exogenous Ghrelin administered either into the ventricles (icv injection) or the arcuate nucleus injection on pain and periaqueducal grey matter (PAG) levels of beta-endorphine and met-enkephaline in rats induced by formalin. The formalin test was used to evaluate pain whereas intracerebral microdialysis was used to monitor opioidergic neuropeptides in the PAG. They show that the injection of ghrelin, whatever the site of injection, counteracted pain and enhanced the levels of opioidergic peptides in the PAG. They conclude that ghrelin has an anti-nociceptive action by acting in the brain, at least in the PAG, on the activity of central, analgesic opioidergic peptides.

The article is interesting and correctly written. The statistical analysis appears to be sound. The figures are also correct (maybe fig 1 could be removed; see below). The article needs additional methodological and technical precisions. Maybe the discussion is not sufficiently close to the actual data reported even it remains connected to the general topic.

1. Ghrelin has been centrally administered. The authors should discuss and/or report in the methods the reason why they choose the dose for icv and intracerebral injections. The absence of dose-response and antagonism experiment limits the interpretation. Thus, the authors should better discuss this methodological aspect.

2.  What would have been the action of the authors if the sham group was different from the other groups? It would change nothing, simply because the sham group cannot be compared to the injected groups with ghrelin. In brief, the authors should remove the sham group from the study. At least, it does not justify a figure.

3. Monitoring neuropeptides using microdialysis is not common and it is often related to the low concentrations present in the dialysates. The concentrations of the opioidergic peptides in the sample seem very elevated. Was it expected? A discussion on this aspect is needed.

4. The authors omitted to report the features of probe cannula, guide-cannula and the providers. Thus, the reader doesn’t know the material of the probe membrane and the cut-off, an important point when dealing with peptides.

5. What the in vitro recovery for the two peptides was? Were the reported results corrected for probe recovery? In fact, the basal values for the two peptides are similar, but beta-endorphine is larger than met-enk (30 amino acids instead of 5 for met-enk) perhaps limiting a good recovery through the probe membrane.

6. The effects of ghrelin are very similar on met enk and beta-endorphine in terms of time course, extracellular concentrations, and the extremely low variability of the effects. It seems that the authors postulate that both are released by pomc expressing neurons. Met enk comes from beta-endorphine structure if we consider only pomc. The authors could discuss a little bit more the neuronal origin of the extracellular levels of these two interdependent peptides (again, if we consider only pomc for met-enk).

7. Cosmetic changes: In the introduction, which is fine, the authors could write before the last paragraph their hypothesis, something like: “Taken these data together, we hypothesized that the brain action of ghrelin disrupts nociceptive influx by enhancing the extracellular concentrations of opioidergic peptides in the PAG”.

8. The neurochemical effects (extracellular levels) induced by ghrelin lasted longer than the behavioral effects. Could the authors discuss these distinct time courses of effects?

Author Response

Many thanks to the Referee for the suggestions. Please find the answer to the different questions

In this article, Pirzadeh et al were interested by the anti-nociceptive effects of Ghrelin and its central effects. They report the effects of exogenous Ghrelin administered either into the ventricles (icv injection) or the arcuate nucleus injection on pain and periaqueducal grey matter (PAG) levels of beta-endorphine and met-enkephaline in rats induced by formalin. The formalin test was used to evaluate pain whereas intracerebral microdialysis was used to monitor opioidergic neuropeptides in the PAG. They show that the injection of ghrelin, whatever the site of injection, counteracted pain and enhanced the levels of opioidergic peptides in the PAG. They conclude that ghrelin has an anti-nociceptive action by acting in the brain, at least in the PAG, on the activity of central, analgesic opioidergic peptides.

The article is interesting and correctly written. The statistical analysis appears to be sound. The figures are also correct (maybe fig 1 could be removed; see below). The article needs additional methodological and technical precisions. Maybe the discussion is not sufficiently close to the actual data reported even it remains connected to the general topic.

1.     Ghrelin has been centrally administered. The authors should discuss and/or report in the methods the reason why they choose the dose for icv and intracerebral injections. The absence of dose-response and antagonism experiment limits the interpretation. Thus, the authors should better discuss this methodological aspect.

The dose was chosen referring to previous experiments. Our study focus on the effect of ghrelin on pain not the comparison of different dose of ghrelin on it.

2.     What would have been the action of the authors if the sham group was different from the other groups? It would change nothing, simply because the sham group cannot be compared to the injected groups with ghrelin. In brief, the authors should remove the sham group from the study. At least, it does not justify a figure.

We used sham group because we wanted to remove the effects of injection stress. Because the injection would have affected met-encephalin or beta-endorphin release. In this way, we have understood that there was no difference between this group and control.

3.     Monitoring neuropeptides using microdialysis is not common and it is often related to the low concentrations present in the dialysates. The concentrations of the opioidergic peptides in the sample seem very elevated. Was it expected? A discussion on this aspect is needed.

The concentration was according to the met-enkepalin and beta-endorphin standards and the samples were collected in 15 minutes. Results are in line with our previous experience.

4.     The authors omitted to report the features of probe cannula, guide-cannula and the providers. Thus, the reader doesn’t know the material of the probe membrane and the cut-off, an important point when dealing with peptides.

4- Cannula: Nidel gage 22, Iran needle company

Length: 3.8mm ICV, 10.2 mm  ARC

Because DV for ICV was 3.6 and for ARC was 10 mm respectively.

Probe: Concentric microdialysis probes with an active dialysis length of 1 mm were constructed in our laboratory from regenerated cellulose dialysis tubing (spectra/pro hollow fiber: molecular weight cutoff: 6000 Da; 0.250 mm OD (Sharp and Zetterstrom, 1992).

5.     What the in vitro recovery for the two peptides was? Were the reported results corrected for probe recovery? In fact, the basal values for the two peptides are similar, but beta-endorphine is larger than met-enk (30 amino acids instead of 5 for met-enk) perhaps limiting a good recovery through the probe membrane.

We saw these results in vitro. Recovery through the probe membrane was checked via study of weight of two peptide and probe membrane (maybe the ability of probe was effective).

6.     The effects of ghrelin are very similar on met enk and beta-endorphine in terms of time course, extracellular concentrations, and the extremely low variability of the effects. It seems that the authors postulate that both are released by pomc expressing neurons. Met enk comes from beta-endorphine structure if we consider only pomc. The authors could discuss a little bit more the neuronal origin of the extracellular levels of these two interdependent peptides (again, if we consider only pomc for met-enk).

Thanks for the suggestion. At present we prefer do not change the Discussion, since the origin of these peptides is not the main topic of the study.

7. Cosmetic changes: In the introduction, which is fine, the authors could write before the last paragraph their hypothesis, something like: “Taken these data together, we hypothesized that the brain action of ghrelin disrupts nociceptive influx by enhancing the extracellular concentrations of opioidergic peptides in the PAG”.

Done. Thanks for the suggestion

8. The neurochemical effects (extracellular levels) induced by ghrelin lasted longer than the behavioral effects. Could the authors discuss these distinct time courses of effects?

Ghrelin was only one of the mediators. It maybe there were others mediators act with injection of ghrelin but we did not measure them.

 Reviewer 2 Report

This manuscript by Pirzadeh et al describes the effects of ghrelin injection into the ARC and lateral ventricle in rats using formalin test, as model of inflammatory pain. Using HPLC analysis from microdialysed samples, they show increased MENK and β-EP concentrations in the PAG. The research design was rigorous, and the statistical analyses used were appropriate to answer the research question.

Author Response

Thanks!

This manuscript is a resubmission of an earlier submission. The following is a list of the peer review reports and author responses from that submission.

Round  1

Reviewer 1 Report

The GHSR1a antagonist used by the authors may have some off-targets effects. The authors should repeat the experiments with a more specific drug like JMV2959. Alternatively, if this is not possible, they should acknowledge the off-target effects and suggest future experiments with more selective blockers.

It is unclear why LEK was not measured. Measurements of LEK should be provided. Alternatively, if this is not possible, a justification should be provided why LEK was not measured.

Did the histology confirm corrected positioning of the cannula in all rats? If not, were rats with no correct positioning excluded from the analysis?

GHSR1a blockade has been proposed as a novel treatment for addictions, including alcohol, cocaine and opioids. For a recent review, see e.g. Zallar LJ, et al. 2017. Here, GHSR1a agonism (or the peptide itself in the experiments) reduces pain. Given the overlap between pain and opioids addiction, how do the authors reconcile this apparent contradiction? The authors should discuss this aspect from both a pharmacological and clinical standpoint 

Author Response

The GHSR1a antagonist used by the authors may have some off-targets effects. The authors should repeat the experiments with a more specific drug like JMV2959. Alternatively, if this is not possible, they should acknowledge the off-target effects and suggest future experiments with more selective blockers.

Although at this time we are unable to repeat the experiments with JMV2959, we have included in the Discussion (p. 8) the possibility of future experiments with more selective blockers such as JMV2959: “It is possible, however, that this GHS-R1α receptor antagonist also has off-target effects, and it would be interesting to carry out further studies with more selective blockers such as JMV2959.”

It is unclear why LEK was not measured. Measurements of LEK should be provided. Alternatively, if this is not possible, a justification should be provided why LEK was not measured.

Leu-enkephalin is an endogenous agonist for the receptors stimulated by opiate alkaloids. It has multiple effects on the CNS, including the neuroendocrine hypothalamus. Leu-enkephalin also controls gonadal function. Met-enkephalin (MENK) is involved in phenomena associated with modulated pain perception, regulation of memory and emotional conditions, food and liquid consumption, and immune system regulation.

In most studies, researchers have investigated met-enkephalin, for example: The analgesic effect induced by microinjection of morphine into the nucleus accumbens can be blocked by naloxone or met-enkephalin antibodies administrated to the PAG, suggesting the existence of the descending pain modulatory pathway from nucleus accumbens to PAG utilizing endogenous opioids, especially with MENK as neurotransmitter (Hjan et al 1986).

Hence, it would not have made sense to investigate leu-enkephalin in the present study.

Did the histology confirm corrected positioning of the cannula in all rats? If not, were rats with no correct positioning excluded from the analysis?

Yes, the histological analysis confirmed the correct positioning of the cannulae and microdialysis probes in all the rats. Modified sentence in Materials and Methods, p. 11:

“The correct position of the cannulae and probe tracing were verified in all the rats by comparison with the rat brain atlas [42].”

GHSR1a blockade has been proposed as a novel treatment for addictions, including alcohol, cocaine and opioids. For a recent review, see e.g. Zallar LJ, et al. 2017. Here, GHSR1a agonism (or the peptide itself in the experiments) reduces pain. Given the overlap between pain and opioids addiction, how do the authors reconcile this apparent contradiction? The authors should discuss this aspect from both a pharmacological and clinical standpoint

Our research on pain is different from that reported in the Zallar review on substance abuse and related behaviors, and we think that there is little relationship between them. We found that ghrelin decreases pain via GHSR1α and opioid secretion.

Reviewer 2 Report

This study investigates the effect of intracerebroventricular or intra-arcuate nucleus administration of ghrelin on formalin-induced pain.  The results show that the antinociceptive effects on ICV administered ghrelin is mediated through the GHS-R1a receptor, whereas in the ARC the effects are mediated through a different ghrelin target. Ghrelin injection in both sites increases beta-endorphin and met-enkephalin concentrations in the PAG. This is an interesting study, but could be improved with some revision.

The results include very detailed descriptions of the ANOVA analysis, which makes it difficult to read. It would be easier if the detail was put into a table and the effects were reported in the results. Discussing the time factor is also confusing in this analysis, since there are different phases to the formalin response.

The abstract suggests that ghrelin and the antagonist were injected independently, but the results say that the antagonist was injected with ghrelin.

The use of MEK as an abbreviation for met-enkephalin is confusing, as it usually refers to MAPK/ERK kinase. It switches between MEK and MK in parts.

All figures: The time should be shown in minutes, rather than 1-12 groups of 5 mins, or samples 1-8. This would make it a lot easier to follow and to compare the beta-endorphin and met-enkephalin levels over time with pain responses.

Figure 1 – Can the large error in the sham group be explained? It looks as there was no intermediate period in this group. Were the n’s for each group the same? The methods say there were 7 groups, n = 7. Does that mean 7 groups of 7, or just 7 groups?

Pg. 3: “…indicating the antagonists lack of effect”: This data suggests that grehlin is acting through a different receptor, rather than the antagonist not working.

Figure 2 – “pain score (n)”: the (n) is not required here as this is a score, not a count of any specific activity.

Figure 2 – The antagonist increases pain at the 15 minute mark – is this expected? Does this suggest that the antagonist is blocking the effect of other mediators in this region? Also, in the discussion third paragraph, pg 18, the authors state that the analgesic effects of grehlin are mediated through the GHS-R1a receptor, but the effects of the antagonist suggests there is an effect on something else. This needs to be addressed.

Methods, pg 9: The groups could be put in a table to make it easier to read.

Pg 10, Section 4.4: “…perfused by means of a microinjection pump into the microdialysis probe with a flow rate of 1.0ml/min” for 15 mins. This is a very large volume!

Pg 10, Section 4.5: “The second phase… ends on the 60th minute”: The data shows no decrease in pain by the 60th minute.

Author Response

This study investigates the effect of intracerebroventricular or intra-arcuate nucleus administration of ghrelin on formalin-induced pain. The results show that the antinociceptive effects on ICV administered ghrelin is mediated through the GHS-R1a receptor, whereas in the ARC the effects are mediated through a different ghrelin target. Ghrelin injection in both sites increases beta-endorphin and met-enkephalin concentrations in the PAG. This is an interesting study, but could be improved with some revision.

No, the ghrelin receptors in the ICV and ARC are likely the same (GHSR-1α). (“It is supposed that ghrelin injection into the lateral ventricle acts via the same receptors as in the ARC albeit with a somewhat stronger effect.”, p. 8)

The results include very detailed descriptions of the ANOVA analysis, which makes it difficult to read. It would be easier if the detail was put into a table and the effects were reported in the results. Discussing the time factor is also confusing in this analysis, since there are different phases to the formalin response.

We prefer to leave the test as it is.

The abstract suggests that ghrelin and the antagonist were injected independently, but the results say that the antagonist was injected with ghrelin.

We injected the ghrelin antagonist first and then ghrelin 10 min later (as described in Materials and Methods, page 9). In the Discussion we called this “co-administration”.

The use of MEK as an abbreviation for met-enkephalin is confusing, as it usually refers to MAPK/ERK kinase. It switches between MEK and MK in parts.

We have changed the abbreviation to MENK.

All figures: The time should be shown in minutes, rather than 1-12 groups of 5 mins, or samples 1-8. This would make it a lot easier to follow and to compare the beta-endorphin and met-enkephalin levels over time with pain responses.

We prefer to leave this as it is.

Figure 1 – Can the large error in the sham group be explained? It looks as there was no intermediate period in this group.

There were a couple of errors in the previous Figure 1, which have now been corrected.

Were the n’s for each group the same? The methods say there were 7 groups, n = 7. Does that mean 7 groups of 7, or just 7 groups?

The sentence in Materials and Methods (p. 9) was changed to “Animals were then divided into seven groups (n=7 rats per group) depending on the drug and the site of injection”.

Pg. 3: “…indicating the antagonists lack of effect”: This data suggests that grehlin is acting through a different receptor, rather than the antagonist not working.

The phrase has been omitted.

Figure 2 – “pain score (n)”: the (n) is not required here as this is a score, not a count of any specific activity.

This has been corrected.

Figure 2 – The antagonist increases pain at the 15 minute mark – is this expected? Does this suggest that the antagonist is blocking the effect of other mediators in this region? Also, in the discussion third paragraph, pg 18, the authors state that the analgesic effects of grehlin are mediated through the GHS-R1a receptor, but the effects of the antagonist suggests there is an effect on something else. This needs to be addressed.

The inhibition generally present between the first and second phase of the formalin test is the result of complex interactions among nociceptive inputs and the descending modulation. At present, we are unable to define at which level this interaction is modulated by ghrelin and/or its antagonist. Further research is necessary to define such interactions.

Methods, pg 9: The groups could be put in a table to make it easier to read.

We do not agree that another table would aid in comprehension. However, we have simplified the description and numbering of the groups.

Pg 10, Section 4.4: “…perfused by means of a microinjection pump into the microdialysis probe with a flow rate of 1.0ml/min” for 15 mins. This is a very large volume!

In fact, the flow rate stated in the text was erroneous and should have been “2.0 µl/min for 15 minutes”. This has been corrected.

Pg 10, Section 4.5: “The second phase… ends on the 60th minute”: The data shows no decrease in pain by the 60th minute.

The pain increase in the second phase (20th-60th min) is related to the release of inflammatory mediators after formalin injection in the hind paw and the pain may continue beyond the end of the formalin test (60th min). Moreover, the analgesic effect of ghrelin is expected to decrease as the ghrelin concentration decreases (according to its half-life).

Round  2

Reviewer 1 Report

.

Reviewer 2 Report

Suggestions to improve this manuscript haven't been addressed and I have serious concerns about the scientific integrity, for example: data points being removed from figure 1 to make the error bars smaller. 

It is now clear from the reviewers responses that the methods are not adequately described or are misleading (eg: co-administration of antagonist-agonist, which are not co-administered according to this response).

This study has too many flaws and the conclusions are not well supported. Some of the suggested changes would have made the manuscript clearer and easier to follow, but they have been ignored.